# Spectrally blue hydrated parent body of asteroid (162173) Ryugu

Eri Tatsumi [1,2,3 ✉], Naoya Sakatani [4], Lucie Riu [5], Moe Matsuoka [5], Rie Honda[6], Tomokatsu Morota [3], Shingo Kameda [4], Tomoki Nakamura[7], Michael Zolensky[8], Rosario Brunetto [9], Takahiro Hiroi [10], Sho Sasaki[11], Sei'ichiro Watanabe [12], Satoshi Tanaka[5,13], Jun Takita[14], Cédric Pilorget[9], Julia de León [1,2], Marcel Popescu[15], Juan Luis Rizos [1,2], Javier Licandro[1,2], Ernesto Palomba[16], Deborah Domingue[17], Faith Vilas[17], Humberto Campins[18], Yuichiro Cho [3], Kazuo Yoshioka [3], Hirotaka Sawada[5], Yasuhiro Yokota[5], Masahiko Hayakawa[5], Manabu Yamada [19], Toru Kouyama [20], Hidehiko Suzuki[21], Chikatoshi Honda[22], Kazunori Ogawa [5], Kohei Kitazato [22], Naru Hirata [22], Naoyuki Hirata[23], Yuichi Tsuda[5,13], Makoto Yoshikawa[5,13], Takanao Saiki[5], Fuyuto Terui[5], Satoru Nakazawa [5], Yuto Takei [5], Hiroshi Takeuchi [5,13], Yukio Yamamoto [5,13], Tatsuaki Okada [3,5], Yuri Shimaki [5], Kei Shirai[23] & Seiji Sugita [3,19]

Ryugu is a carbonaceous rubble-pile asteroid visited by the Hayabusa2 spacecraft. Small rubble pile asteroids record the thermal evolution of their much larger parent bodies. However, recent space weathering and/or solar heating create ambiguities between the uppermost layer observable by remote-sensing and the pristine material from the parent body. Hayabusa2 remote-sensing observations find that on the asteroid (162173) Ryugu both north and south pole regions preserve the material least processed by space weathering, which is spectrally blue carbonaceous chondritic material with a 0–3% deep 0.7-μm band absorption, indicative of Fe-bearing phyllosilicates. Here we report that spectrally blue Ryugu's parent body experienced intensive aqueous alteration and subsequent thermal metamorphism at 570–670 K (300–400 °C), suggesting that Ryugu's parent body was heated by radioactive decay of short-lived radionuclides possibly because of its early formation 2–2.5 Ma. The samples being brought to Earth by Hayabusa2 will give us our first insights into this epoch in solar system history.

[1] Instituto de Astrofísica de Canarias (IAC), La Laguna, Tenerife, Spain. [2] Department of Astrophysics, University of La Laguna, La Laguna, Tenerife, Spain. [3] The University of Tokyo, Bunkyo, Tokyo, Japan. [4] Rikkyo University, Toshima, Tokyo, Japan. [5] Institute of Space and Astronautical Science (ISAS), Japan Aerospace Exploration Agency (JAXA), Sagamihara, Kanagawa, Japan. [6] Kochi University, Kochi, Kochi, Japan. [7] Tohoku University, Sendai, Miyagi, Japan. [8] NASA Johnson Space Center, Houston, TX, USA. [9] Université Paris-Saclay, CNRS, Institut d'Astrophysique Spatiale, Orsay, France. [10] Brown University, Providence, RI, USA. [11] Osaka University, Toyonaka, Osaka, Japan. [12] Nagoya University, Nagoya, Aichi, Japan. [13] SOKENDAI (The Graduate University for Advanced Studies), Hayama, Kanagawa, Japan. [14] Hokkaido Kitami Hokuto High School, Kitami, Hokkaido, Japan. [15] Astronomical Institute of the Romanian Academy, Bucharest, Romania. [16] NAF, Instituto di Astrofisica e Planetologia Spaziali, Rome, Italy. [17] Planetary Science Institute, Tucson, AZ, USA. [18] University of Central Florida, Orland, FL, USA. [19] Planetary Exploration Research Center (PERC), Chiba Institute of Technology, Narashino, Chiba, Japan. [20] National Institute of Advanced Industrial Science and Technology (AIST), Koto, Tokyo, Japan. [21] Meiji University, Kawasaki, Kanagawa, Japan. [22] The University of Aizu, Aizu-Wakamatsu, Fukushima, Japan. [23] Kobe University, Kobe, Hyogo, Japan. ✉email: etatsumi-ext@iac.es

Our understanding of the evolution of dust to planetesimals in the early solar system is advancing thanks to the integration of intensive spacecraft exploration, dynamical modeling, meteoritic analyses and advances in astronomical observations. Asteroids are considered to be remnants of planetesimals, which were the main building blocks of terrestrial planets. Thus, the thermal history and water/rock ratio of asteroids are keys to understanding the physical and chemical environments in the early solar system. The NASA Dawn spacecraft investigated two protoplanets that have survived almost intact, (1) Ceres and (4) Vesta. Geomorphological and spectral analyses have recently suggested the presence of liquids in the subsurface of Ceres[1]. Thus, carbonaceous protoplanets have probably undergone rock-ice differentiation and/or varying degrees of internal aqueous alteration[2]. The rubble-pile asteroid Ryugu, target of the Hayabusa2 mission, resulted from catastrophic disruptions of larger parent bodies[3–5]. Ryugu could be an ideal body to study the thermal history and water/rock ratio of pre-disruption parent bodies, much larger (~100 km) than Ryugu. Our objective is to find the most pristine material on Ryugu and to study evidence for proposed thermal metamorphism of its original parent body and consequent processes after its catastrophic disruption and re-accumulation. Stratigraphic analyses have suggested that the possible surviving, unprocessed materials are bluer than the average Ryugu reflectance spectrum[3,6]. Slightly bluer materials than the average were found on the equatorial ridge, which might be uncovered by regolith migration from the ridge to the middle latitudinal regions[3,7]. Moreover, the global observations obtained by the Hayabusa2 spacecraft discovered that the bluest materials are distributed at the polar regions of Ryugu, where both solar heating and space weathering are weakest[3,5] (Supplementary Fig. 1). This motivated us to conduct detailed surveys of the polar regions to investigate the unprocessed materials. Effects after the formation of Ryugu, such as solar wind irradiation, micrometeoroid bombardment, and radiative heating caused by close encounter to the Sun, need to be deconstructed. Phyllosilicates in CM (Mighei-type)/CI (Ivuna-type) chondrites become progressively enriched in Mg (and depleted in Fe) as aqueous alteration proceeds[8–10]. Thus, Fe-bearing phyllosilicates showing 0.7-μm band absorption are a strong indication of the specific water/rock ratio condition during the parent body formation. OH-band absorption around 2.7 μm also provides hints for hydration-dehydration states. The peak position of OH-band evolves from 2.8 to 2.7 μm, attributed to Fe-rich to Mg-rich phases of phyllosilicates[11].

In this study, we reveal the hydrated state of the main components of Ryugu, which are inherited from its parent body, based on remote-sensing observations. Here, we show the least processed materials by space weathering in Ryugu's polar regions, which are spectrally blue and associated with 0.7-μm band absorption. This finding suggests intensive aqueous alteration and subsequent heating up to 670 K (400 °C) of Ryugu's parent body.

## Results

**Remote sensing observations.** The Telescopic Optical Navigation Camera (ONC-T) is a multi-band imager that is equipped with seven band filters ranging wavelength from 0.40 to 0.95 μm[12]. Figure 1 shows the spectral index images of both poles; visible spectral slope (b–x; 0.48 μm to 0.86 μm) and 0.7-μm band absorption (see "Methods", subsection Image analyses). The 0.7-μm band absorption is attributed to $Fe^{2+}$–$Fe^{3+}$ charge transfer of oxidized iron[13] and is often used as a proxy for phyllosilicates such as serpentine and saponite. Local variations in spectral slope and 0.7-μm band absorption were observed. Spectrally blue (negative visible spectral slope) material is concentrated on both

poles, as clearly shown (See also Supplementary Fig. 1 and Supplementary Movies 1 and 2). Furthermore, blue material is associated with a relatively deeper 0.7-μm band absorption (Supplementary Fig. 2). Otohime Saxum, the largest boulder on the south pole, is of particular interest in this regard (Fig. 1b–d). As was pointed out in the global Ryugu observations, Otohime has multiple sharp facets on its surface and many cracks that could have been caused by meteorite impacts[3]. Furthermore, Otohime consists of three facets, two flat facets (A and B) and one rough facet (C) (detailed regions of interest for spectral properties in Fig. 2 are shown in Supplementary Fig. 4). On the south pole, the blue Facets A and B show a relatively deeper 0.7-μm band absorption than Facet C, which has a spectral slope similar to the average for Ryugu. The spectra of north pole and Otohime's Facet A exhibit the bluest spectrum (b–x slope of −0.17/μm and −0.14/μm, respectively) and a relatively deeper 0.7-μm band absorption (Fig. 2). The depth of 0.7-μm absorption for the north pole region and Facet A are 1.24 ± 0.11% and 1.28 ± 0.06%, respectively, while the typical Ryugu and Facet C absorption values are 0.98% and 0.63 ± 0.08%. This absorption is much deeper than the hint of absorption excess of ~0.07% at low lattitudes[14]. Because the ambiguity of the 0.7-μm absorption measurement from the sensitivity calibration of ONC-T is 1.6%[12], the absolute value of the 0.7-μm band absorption of the north pole region and Facet A is less than this ambiguity. However, because this ambiguity is dominated by the spectral uncertainty of standard stars, the accuracy of spectral difference between different areas on Ryugu is much better, ~0.1% for 15 pixel by 15 pixel binning[12]. Thus, the depth of 0.7-μm band is statistically significant, ~3–7 sigma (see "Methods", subsection Image analyses).

The near-infrared reflectance spectrometer (NIRS3)[6] observations display a 2.72-μm band absorption in the spectrum of the north pole plains that shows a slightly deeper absorption than the standard reference spectrum (8.1% on the northern plain and 7.5% for the reference spectrum; see Fig. 3a). The increase in band depth is small, however, when comparing the band depth for all NIRS3 spectra observed on 26 July with the spectra falling on the northern plain, we do observe a region-wide increase in the band depth (Fig. 3a, right panel). Facet A of Otohime shows a slightly larger band depth (8.85% on Facet A and 8.35% for the reference spectrum, see Fig. 3b), however the spectra that fall on Otohime for this observation display a large variation of the 2.72-μm feature. It is thus difficult to understand whether this increase is significant. Facet B of Otohime shows no increase in the band depth (7.7% on Facet B and 7.9% for the reference spectrum, see Fig. 3c). Overall, a slight decrease is observed for this facet (Fig. 3c, right panel). Facet C of Otohime shows no significant changes in the band depth compared with the standard reference averaged spectrum (8.5% on Facet C and 8.3% for the reference spectrum, see Fig. 3d). In contrast with the Small Carry-on Impactor (SCI) crater observation[15], no peak shift has been observed in the spectra for the polar regions.

It is known that the 0.7-μm and 2.7-μm features can be easily deformed by either heating or space weathering[16–18]. Even though the absolute value of the 0.7-μm band depth has formal uncertainties ranging from 0 to 3%, the strong correlation with the high latitude regions suggests the presence of the 0.7-μm band absorption is real. Given that the blue materials show slightly deeper 0.7-μm band absorptions on the polar regions where they are less irradiated by the Sun, we anticipate that heating or space weathering decreases the absorption features.

**Solar heating and solar wind irradiation.** To examine the cause of the distribution of blue materials with 0.7-μm band absorption, the maximum temperature in an asteroidal year and the normalized

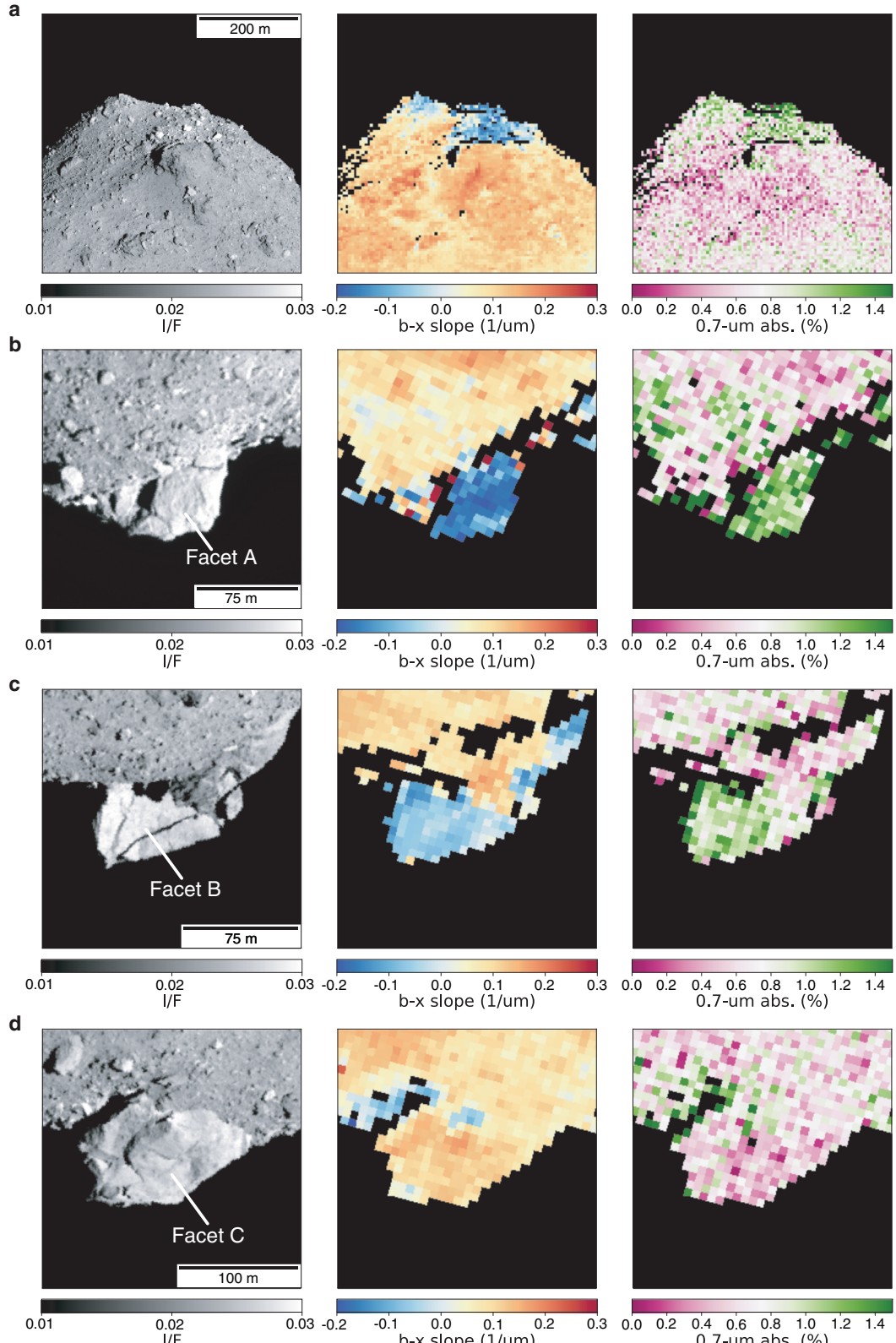

**Fig. 1 I/F and color index maps of the both polar regions. a** North pole. **b–d** Otohime Saxum at south pole (**b**: Facet A, **c**: Facet B, **d**: Facet C); (Left) radiance factor (I/F) images, (Middle) b–x (0.48–0.86 μm) spectral slope map, (Right) 0.7-μm absorption depth map. Image ids used for this figure are listed in Supplementary Table 1.

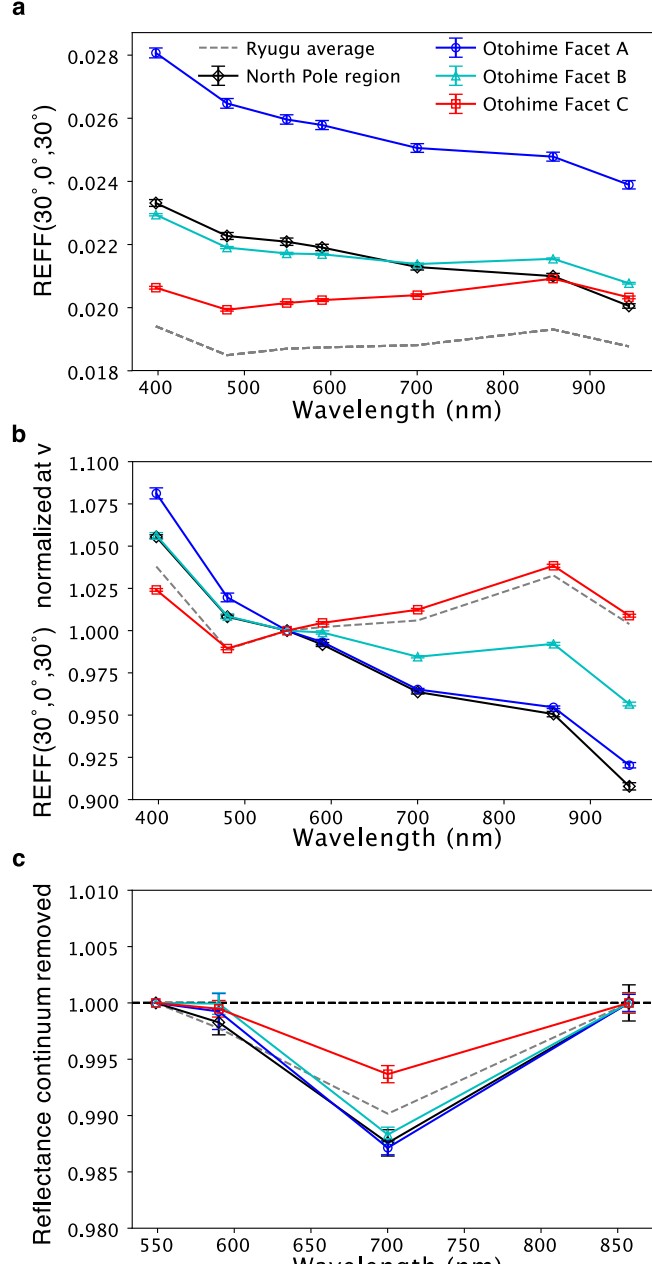

**Fig. 2 Spectrophotometric characteristics of Otohime's different facets and the north polar region.** The region of interests are referred in Fig. 1. Standard error in the region of interests (Supplementary Fig. 1) are shown as error bars. **a** Absolute value of the reflectance factor at (*i, e, α*) = (30°, 0°, 30°). **b** Normalized spectra at v band of (**a**). **c** Continuum removed spectra over the v to x band range.

solar photon dose were calculated based on the shape model and current orbital elements (Figs. 4 and 5, see "Methods", subsection Temperature and photon dose calculation). It should be noted that micrometeoroid bombardment can contribute similarly to the solar photon dose distribution because micrometeoroids are perturbed by radiation, solar wind pressure, and Poynting-Robertson drag, which are radial directional effects[19]. Thus, in this study, we define space weathering effects as either by solar wind irradiation or micrometeoroid bombardment. Although in the current orbit the surface temperature cannot reach 570 K, where decomposition of phyllosilicates[20] can occur, Ryugu once orbited closer to the Sun[7].

For example, at the heliocentric distance of 0.4 au, the maximum temperature of a large part of the surface could reach >570 K and the north polar region was kept <570 K (Supplementary Fig. 3). Figure 6 compares the observational values obtained by ONC-T and simulated values from models of solar wind irradiation and heating by the Sun. The comparison indicates that areas and facets with low solar wind irradiation fluxes exhibit blue spectra and relatively deeper 0.7-μm absorption, while the influence of solar heating does not clearly correlate with the spectral characteristics. On the north pole, the concentration of the material with blue spectral slope, and deeper 0.7-μm band absorption were well correlated with the regions experiencing the lowest temperatures and least solar wind irradiation due to the large incidence angle during the whole asteroidal year. Furthermore, this location is shadowed by the rock just in front of the blue region. On the other hand, at the south pole, Otohime's facets responded differently to both processes. Facet A is affected less by solar irradiation because the hill on the opposite side shields this facet, while this facet could be heated as high as the average temperature due to the low incidence angle to the Sun. Facet C is heated and irradiated as much as the average level. The correspondence between the color variation and those processes shows that solar wind irradiation is a more likely cause for the color changing from blue to red and the decrease in the 0.7-μm band absorption depth (Fig. 6 and Supplementary Movies 1 and 2). On Ryugu we see the non-linear optical effects associated with solar wind irradiation, which are also observed in the laboratory ion irradiation experiments[21]. The correlation with less space weathering at the polar regions was also observed on the Moon by SELENE[21]. Moreover, there is not a significant difference in the 2.7-μm band absorption in the polar regions, suggesting that Ryugu's pristine material still is not rich in hydrous silicates. No significant difference in 2.7-μm is consistent with the laboratory experiments showing that space weathering does not strongly affect the OH-band depth[22]. Even when Facets B and C receive similar solar-wind irradiation, Facet B shows a bluer spectrum, indicating a fresher surface. This may be a result of Facet B's more active resurfacing possibly by thermal fatigue owing to its nearly vertical slope with respect to the geoid.

The solar-wind irradiation dose of Facet A is <20% of that experienced by the average surface. This suggests that Facet A takes more than 5 times longer to be space weathered by the solar wind than typical locations on Ryugu. Because Ryugu's color variation is the ongoing process induced by space weathering, the space weathering timescale for Ryugu's surface is on the order of the surface's exposure time, which was estimated as ~$10^5$ years or younger for top 1 m[23]. Considering that laboratory experiments simulating space weathering show unidirectional change in spectral slope, the original color of materials on Ryugu is commensurate with a B-type in Bus's taxonomy, which is characterized by a negative visible spectral slope[24,25]. This wide range of color variation suggests that the slightly hydrated B-type asteroids can evolve into Cb-type (C/F-type in Tholen's taxonomy)[26], negative to positive spectral slope, by space weathering. Previous experimental studies of hydrated CI and CM carbonaceous chondrites suggested that the color variation by space weathering is sensitive to not only the starting composition but also the sample characteristics (slab vs. pellet vs. loose powder) and radiation beam properties (fluence and energy)[27,28]. So far, ion irradiation experiments on hydrated carbonaceous chondrites, which simulate space weathering by solar wind, have not reproduced the same spectral variation on Ryugu[17]. The fact that we still have not reproduced the exact trend as seen on Ryugu may reflect that we do not have the appropriate starting material yet, suggesting that the returned sample will be somewhat different from what we have in the meteorite collection.

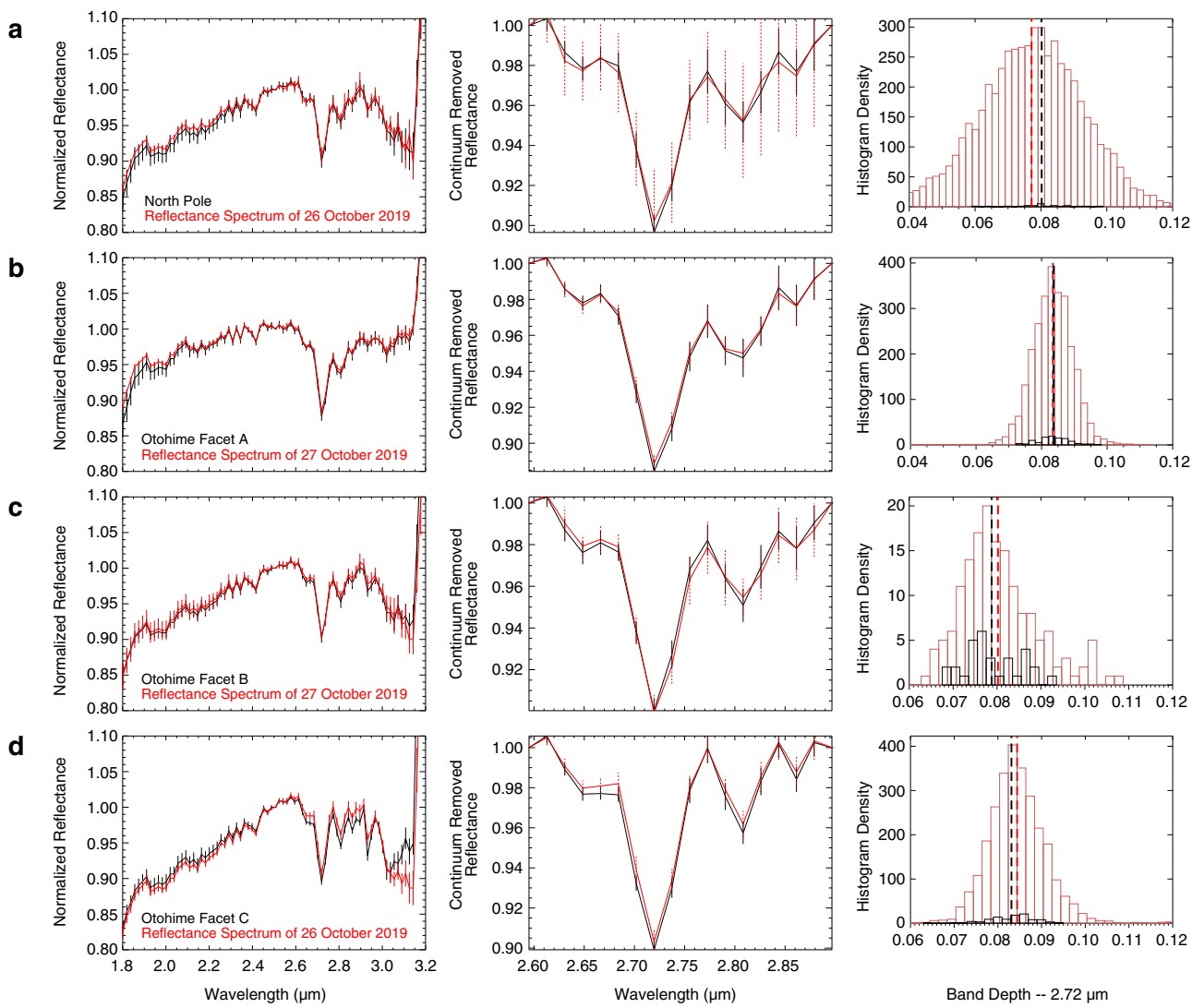

**Fig. 3 Near-infrared observations for polar regions.** (Left) Normalized NIRS3 reflectance spectra of the polar regions compared with the standard reference spectrum outside of the polar regions. The spectra are normalized at 2.5 μm. The error bars are indicative of the standard deviation amongst the spectra used to compute the average region of interest (black) spectrum and average standard reference spectrum (red). (Middle) Detail of the 2.72-μm feature regions. We show the continuum removed band. The continuum is removed using 2.6 μm and 2.9 μm as anchor points[67]. The error bars are indicative of the standard deviation amongst the spectra used to compute the average region of interest (black) spectrum and average standard reference spectrum (red). (Right) Histogram distribution of the band depth for each day's observation. All spectra in the day (red), spectra of the region of interest (black). The vertical dashed lines represent the average value of each distribution. **a** North pole (photometrically corrected data). **b** Otohime Facet A (non-photometrically corrected data). **c** Otohime Facet B (photometrically corrected data). **d** Otohime Facet C (photometrically corrected data).

**Comparison with possible parent bodies and B-type asteroids.** We can also discuss the nature of Ryugu's original parent body based on the color changes induced by space weathering. Our observations indicate that Ryugu has a stronger connection with B-type (including both B and F types in Tholen's taxonomy, the main difference is UV-blue absorption[24]) than C-type asteroids because reddening is a relatively recent event on Ryugu. An asteroid family whose members share similar orbital elements is formed by catastrophic disruption of the original parent body. The observed variation of b-x (0.40–0.86 μm) spectral slope is $-0.17\,\mu m^{-1}$ for the north pole region compared to $0.10$–$0.12\,\mu m^{-1}$ for the global average and Otohime Facet C. The spectral slope range of the inner main belt asteroids in the Eulalia and Polana families are within the same range as the variation observed on Ryugu[29], while the Erigone family members show more variation in the redder range. This

suggests that Ryugu could have originated from either the Polana or Eulalia families. Furthermore, the spectral properties of Otohime's blue facets are similar to those of Polana and Eulalia's blue global average spectra (Fig. 7). The x (0.86 μm) and p (0.95 μm) spectro-photometric values (Fig. 2) highlight a possible feature that can be attributed to olivine-pyroxene mixtures. The comparison with other asteroid spectra (Fig. 7) shows that in the wavelength interval covered by x and p filters, Polana and Eulalia have a similar behavior to Ryugu. In addition, a flat to upturn in the UV-blue region is characteristics of F type in Tholen's taxonomy, which is mainly populated in the inner main belt[24]. Thus, Ryugu might be a representative of those spectrally blue members of inner main belt.

B-type asteroids are common in the near-Earth asteroid population (about 5% based on visible spectra[30]). The upturn in the UV-blue region is a peculiar feature highlighted in the

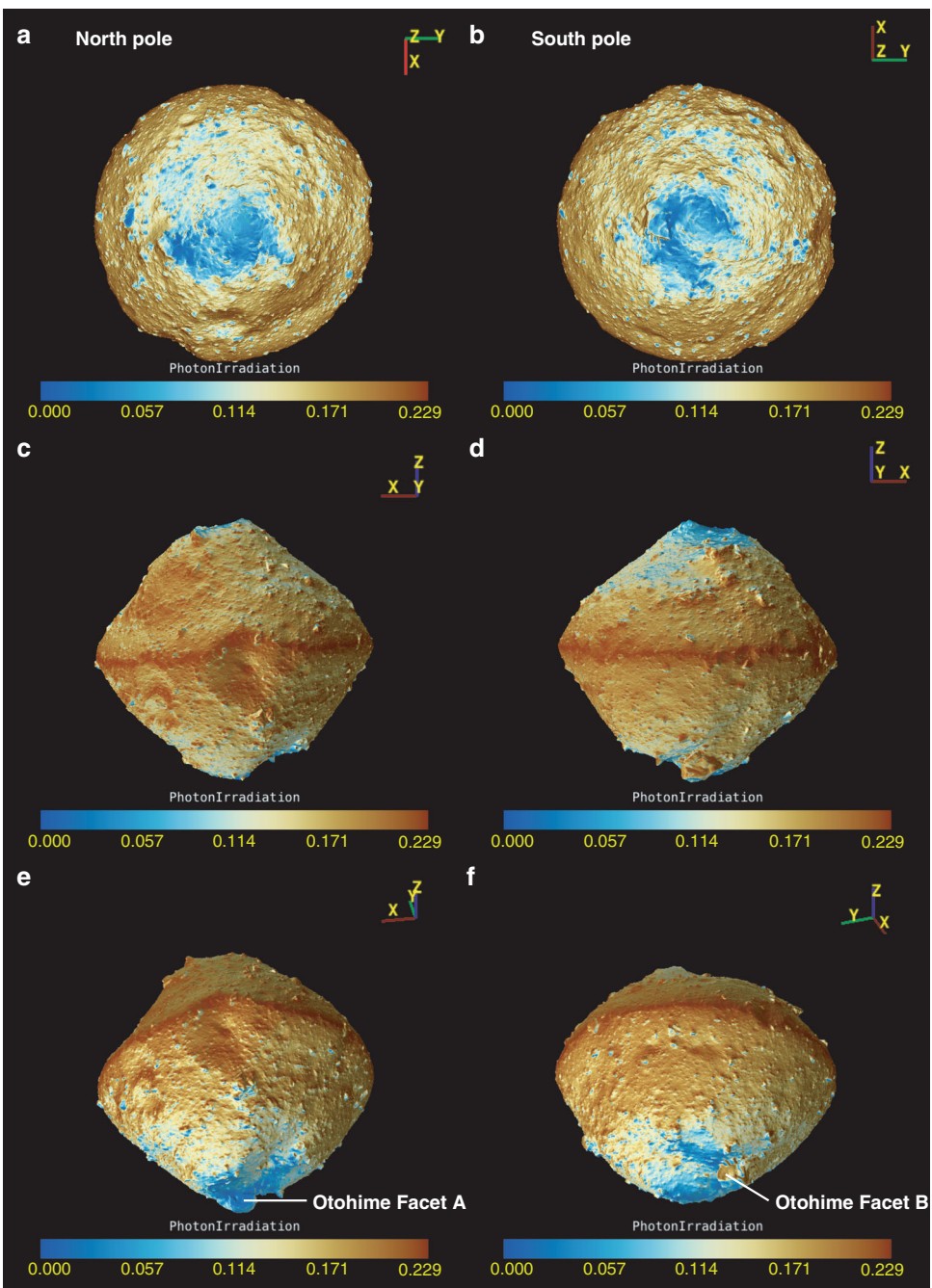

**Fig. 4 Global distribution of solar photon dose averaged over one asteroidal year.** This value is normalized by the photon dose experienced by a facet perpendicular to the Sun at 1 au away. Thus, the color bars show non-dimensional value. **a**, **b** are nadir views from the north and south poles. **c**, **d** are the views from (LAT, LON) = (0°, 90°) and (0°, 270°), respectively. **e**, **f** are the views from (−30°, 100°) and (−30°, 200°) to show Facets A and B of Otohime Saxum. Photon dose distribution shows lower irradiation for the polar regions. Otohime's Facet A shows very low irradiation due to the hill just in front of it.

spectra of (3200) Phaethon[31] and (101955) Bennu[32] (Fig. 7). Bennu is currently being investigated by the OSIRIS-REx spacecraft[33]. We cannot rule out that Bennu and Ryugu originated from the same parent body, considering their similar albedo, bulk density, and thermal inertia[26,33–35]. Although the visible spectrum of Bennu closely matches that of Otohime's blue material, Bennu shows a stronger 2.7-μm OH-band absorption than Ryugu and also a different band shape[32]. It was suggested recently that the differences in degree of hydration can also have been produced by one impact[4]. In contrast, the difference of the

bright exogenic boulders on these two asteroids suggest the different parent bodies at least for one generation[5,36]. Furthermore, our new result suggests different space weathering variations in visible wavelengths: Ryugu shows reddening, while Bennu shows bluing[37]. Also the fact that a region on Ryugu with a similar OH-band feature to Bennu has not been found suggests a lower possibility that they originated from the same parent body.

(3200) Phaethon, the target of the DESTINY + Mission[38], also exhibits blue spectra in the visible wavelength range and turn-up

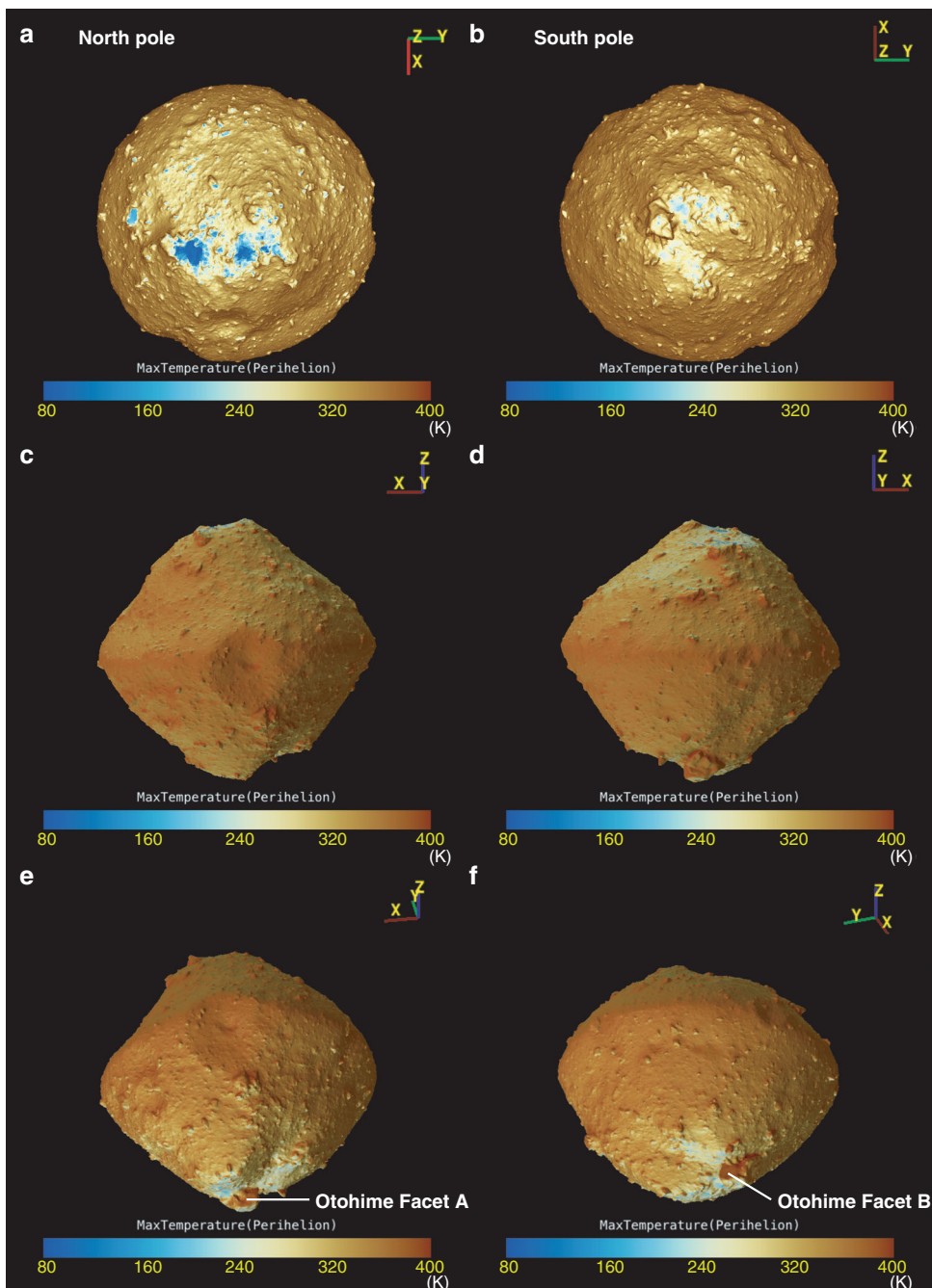

**Fig. 5 Global distribution of maximum surface temperature at perihelion. a**, **b** are nadir views from the north and south poles. **c, d** are the views from (LAT, LON) = (0°, 90°) and (0°, 270°), respectively. **e**, **f** are the views from (−30°, 100°) and (−30°, 200°) to show Facets A and B of Otohime Saxum. Maximum temperature shows low values at the north pole <100 K. However, Otohime shows high surface temperatures even on the pole due to the high incidence angles towards the Sun.

in the UV. Recently, many ground-based observations were made over a wide wavelength range, which reported that the variation in visible spectral slope depends on rotational phase. The range of spectral slope variation in one rotation is −0.5 to 0.05 μm$^{-1}$ and that of the relative R-band magnitude is ±0.15[31,39]. Moreover, a correlation between brighter and bluer spectra was also observed[31]. The similarity for both Ryugu and Phaethon, that neither exhibits a strong UV nor 2.7-μm OH-band absorption for the entire rotational phases[40], suggests similar spectral changes due to space weathering on both asteroids. Thus, the majority of Phaethon's surface could be fresh due to rejuvenation caused by the recent encounter with the Sun, i.e., fresh cometary activity.

## Discussion

The 0.7-μm band absorption for Ryugu is indicative of Fe-bearing phyllosilicates. In contrast, a peak OH-band at 2.72 μm[6] suggests the presence of Mg-rich phyllosilicates. Thus, Ryugu might contain an appreciable amount of both Fe- and Mg-rich phyllosilicates. Although there is no good spectral and physical match to our well-investigated samples in terms of darkness[3], shallow 2.7-μm[6] and 0.7-μm bands, and low thermal inertia[35,41,42], here the hydrothermal history is discussed based on our current knowledge. Nevertheless, it is highly possible that Ryugu's composition contains materials not sampled in our meteorite collections.

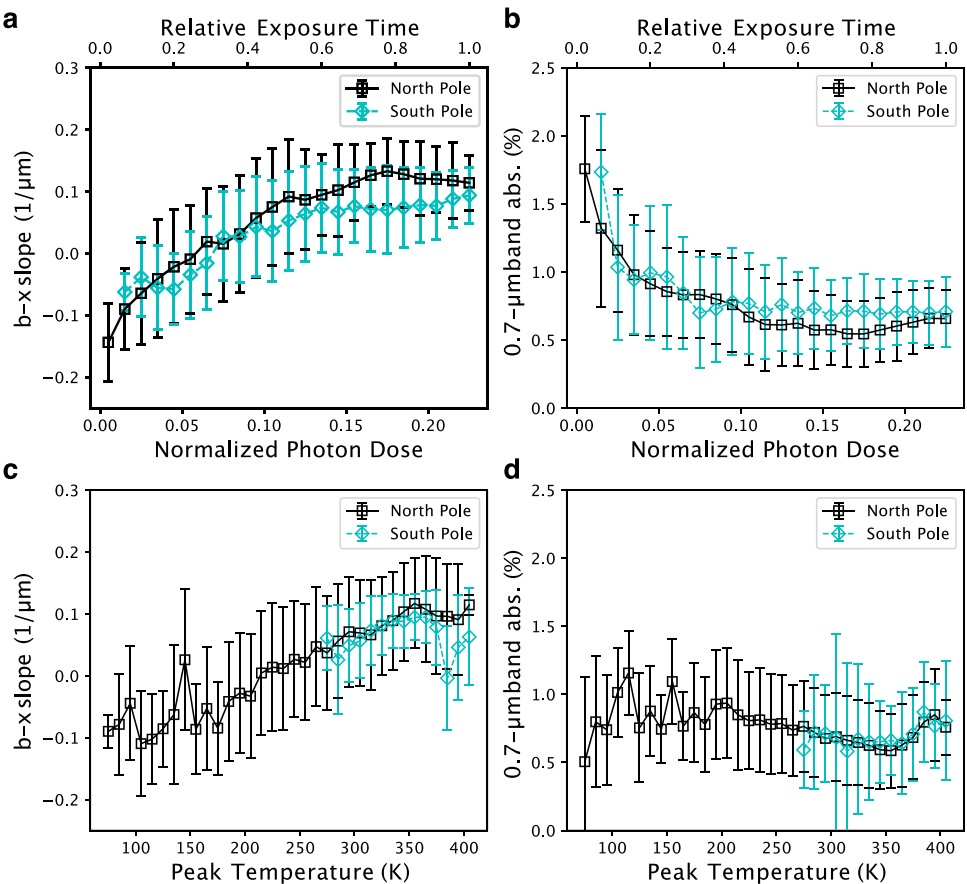

**Fig. 6 Comparison between observed spectrophotometric indexes and modeled values. a** b–x slope value versus normalized photon dose. **b** 0.7-μm band absorption versus normalized photon dose. **c** b–x slope value versus peak temperature at perihelion. **d** 0.7-μm band absorption versus peak temperature at perihelion. The north polar observation (Fig. 1a) and the south pole observation (Fig. 1c) are shown with black and cyan markers, respectively. Observed spectrophotometric indexes are binned by every 0.1 for photon dose and every 10 K for peak temperature. Error bars indicate the standard deviation in each bin. Relative exposure time for **a**, **b** is calculated such that the most irradiated surface is defined as unity. Image ids used for this figure are listed in Supplementary Table 1.

The presence and degree of both 0.7-μm and 2.7-μm features on Ryugu allows us to constrain the parent body thermal history and water abundance. Previous global studies of Ryugu suggest two scenarios for the main composition of Ryugu:[3,6] (1) aqueously altered and thermally metamorphosed carbonaceous chondrite (ATCC including CY) material, (2) incipiently altered inter planetary-dust-particle-like, or cometary material. The 0.7-μm band absorption is closely related to the amount of Fe-rich phyllosilicates[10]. Up to now, the 0.7-μm feature has been detected in spectra of CM chondrites, ATCCs and only one CR chondrite[43,44]. However, the CR chondrites usually display a much higher reflectance[45]. The range of the 0.7-μm band depth for CM chondrites and ATCCs are 0.5–6% and <2%, respectively[10,43,44], which overlaps the observed 0.7-μm band depth of 0–3% on Ryugu. Thus, Ryugu's parent body was likely formed in a similar environment as CM chondrites in terms of water/rock ratio to produce the 0.7-μm band absorption. Nevertheless, a weak OH-band absorption, ~8%, and its short peak wavelength at 2.72 μm is not common among the general CM chondrite feature, although some highly aqueously altered CM chondrites also show a sharp OH-band peak close to 2.7 μm[11,46]. This OH-band feature can be reproduced by a peak shift toward shorter wavelengths by partial dehydration[11,44]. Moreover, thermal metamorphism is also known to decrease the near-ultraviolet absorption as well[16]. Such a mild dehydration

process could have resulted in a flat near-ultraviolet spectrum, a weak 0.7-μm band absorption, and a decreased OH-band absorption with peak position at 2.72 μm. Thus, scenario 1 with the connection to aqueously altered and thermally metamorphosed CM-like material is more plausible. Because phyllosilicates are completely decomposed at 870–970 K (600–700 °C)[47], the maximum temperature that Ryugu's parent body experienced must have been lower than this level. Moreover, laboratory experiments on CM chondrites reported that although the 0.7-μm band absorption due to Fe-rich phyllosilicates drastically decreases at ~570–670 K (300–400 °C)[16,20], these phyllosilicates are usually transformed into highly desiccated and disordered intermediate phases at 570–870 K (300–600 °C)[47–50]. Thus, the heating temperature for Ryugu might be lower than 670 K (400 °C). Otherwise, heterogeneous heating and dehydration might also contribute to the preservation of 0.7-μm band absorption. Besides, the fresh material on Ryugu has a negative slope in the visible wavelength range (0.40–0.95 μm) and a positive slope in the near-infrared wavelength range (1.9–2.6 μm), suggesting a concave shape from visible to near-infrared reflectance spectrum. A weak 0.7-μm feature associated with a broad absorption feature around 1 μm is often observed in the visible to near-infrared spectra of ATCCs and B-type asteroids[43,51]. The concave shape of the spectra around 1 μm can be indicative of either olivine, Fe-bearing phyllosilicates[43], and/or magnetite[51].

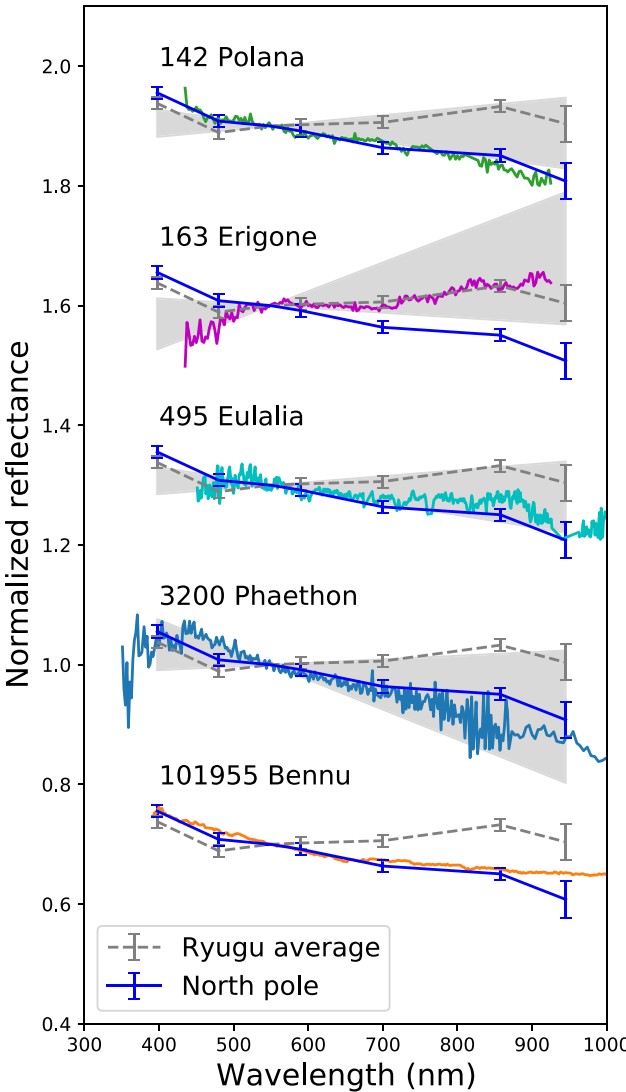

**Fig. 7 Least processed region on Ryugu compared with spectrally blue asteroids.** Spectral comparison of the north polar region (solid blue line) with the possible parent bodies, (142) Polana, (163) Erigone, and (495) Eulalia, the B-type near-Earth asteroid (3200) Phaethon and (101955) Bennu. Spectra are normalized at 0.55 μm and offset by 0.3. Reflectance spectra of (142) Polana, (163) Erigone, (495) Eulalia, (3200) Phaethon, and (101955) Bennu were obtained by refs. [25,68,69], and ref. [32], respectively. Gray hatches for (142) Polana, (163) Erigone, and (495) Eulalia indicate 80% of the distribution of slope among their family members[29,70]. Gray hatch for Phaethon indicates spectral variation over one rotation[31].

Alternatively, a currently unsampled carbonaceous chondrite that experienced intensive hydration and subsequent heating would produce Ryugu's unique spectral features. To summarize, the weak 2.7-μm band absorption and presence of the 0.7-μm feature could be explained by aqueous and thermal alteration of a CM-like precursor (scenario 1), which corresponds to Stage II in the thermal metamorphic stages defined by ref. [50] or an unsampled hydrated carbonaceous chondrite. CYs are not the best candidate for Ryugu's material because they are typically heated >770 K (>500 °C) and are highly dehydrated[52].

Oxygen isotopic compositions suggests that CI chondrites were formed with higher water/rock ratios than CM chondrites[9,53,54]. Our results support that Ryugu's parent body may have formed under similar water/rock ratio conditions as CM chondrites, 0.3–

0.6[53], but subsequently experienced thermal metamorphism to a maximum of 570–670 K (300–400 °C). The uniformity in visible and near-infrared spectra[3,6] implies either that there were no large displacements or differentiation between rock and ice particles in the parent body even at high temperatures, or that dust and water were convecting as a mud ball[55]. However, possible heterogeneity in the parent body has been suggested by the Hayabusa2 Thermal Infrared Imager (TIR)[35,42] and ONC-T measurements in anomalous regions, although the portion of Ryugu covered by these regions is very minor. Very high density and bright materials on Ryugu[5,35] might have been caused by hot pressing occurring over only a minor part of parent body, possibly in its center. Ryugu possesses an extremely porous uppermost layer which displays low reflectance and extremely low thermal inertia[56]. These properties suggest that Ryugu's parent body might have been ~100 km in diameter and had layers with different degrees of thermal metamorphism (Fig. 8). The detailed thermal history will be addressed by returned sample analyses. However, the majority of the asteroid has been highly aqueously altered suggested by the 0.7-μm band absorption and later partially dehydrated by temperatures of up to 570–670 K. The highest temperature achieved by the decay of $^{26}$Al (half-life 0.7 Myr) is sensitive to the accretion age of the parent body. In contrast, heating due to impact is less likely because it would accompany with impact-induced compaction[4], which is inconsistent with Ryugu's highly porous boulder structure observed on Ryugu[35,41]. A planetesimal would have required a relatively high ratio of $^{26}$Al to achieve temperatures sufficiently high to cause thorough dehydration, corresponding to early accretion, 2–2.5 Ma post Calcium-Aluminum-rich Inclusions[2,57], while non-ATCC CM chondrites record low temperature aqueous reactions (typically <420 K), corresponding to 3–4 Ma by isotopic dating[58]. It should be noted that aqueous alteration temperatures among non-ATCC vary over a wide range 250–550 K[59–61]. Besides, ATCCs in the meteorite collections are also reported to have experienced a variety of heating temperatures, 470 K to >1000 K[49,50,61] (Fig. 8). Although several works suggested the thermal metamorphism for these meteorites was likely caused by short-lived events, such as impacts or solar radiations[49,62,63], the heating mechanism is currently poorly constrained yet. Ryugu's samples may be the first example of heating by primordial radiogenic decay, as suggested by its highly porous nature. The wide range of formation temperatures of CM chondrites and Ryugu (250–670 K) provides constraints on the formation conditions of water-rich carbonaceous asteroids in the early solar system. Although the Fe-bearing phyllosilicates of Ryugu suggests a similar water/rock ratio to CM chondrites, the formation timing of Ryugu's B-type parent body, possibly Polana or Eulalia, might be earlier than CM chondrites' parent bodies.

## Methods

**Image analyses.** Global spectral slope in Supplementary Fig. 1 was made based on the global observations at an altitude of 20 km on 12 July, 24 August 2018 and 24 January 2019. The bias correction and the updated flatfield correction was applied based on ref. [14]. The digital count images were converted into radiance factor (I/F) images by the radiometric calibration[12,64]. All the I/F images were photometrically corrected to the standard geometry of (i, e, α) = (30°, 0°, 30°) based on the photometry model by ref. [26] with the shape model of SFM_3M_v20200815 and SPICE kernels (see "Data availability"), where i, e, and α are incidence, emission, and phase angles, respectively. The shape model was derived in the same manner as ref. [34] but using all the image from proximity operations around the asteroid. We used pixels with $i < 70°$, $e ≤ 70°$, and I/F > 0.005 in order not to introduce large error by photometric correction and shadow regions. The b (0.48 μm)- to- x (0.86 μm) spectral slope was calculated as done by ref. [3]. The bluest regions are found in the pole regions.

Following the global observations at an altitude of 20 km, Hayabusa2 conducted closer approaches to the pole region at an altitude of 5 km on 28 February, 1 March, and 26 October 2019. Their phase angles were 17°, 17°, and 34°, respectively. The images were taken by ONC-T with 7-color filters (ul: 0.40 μm, b:

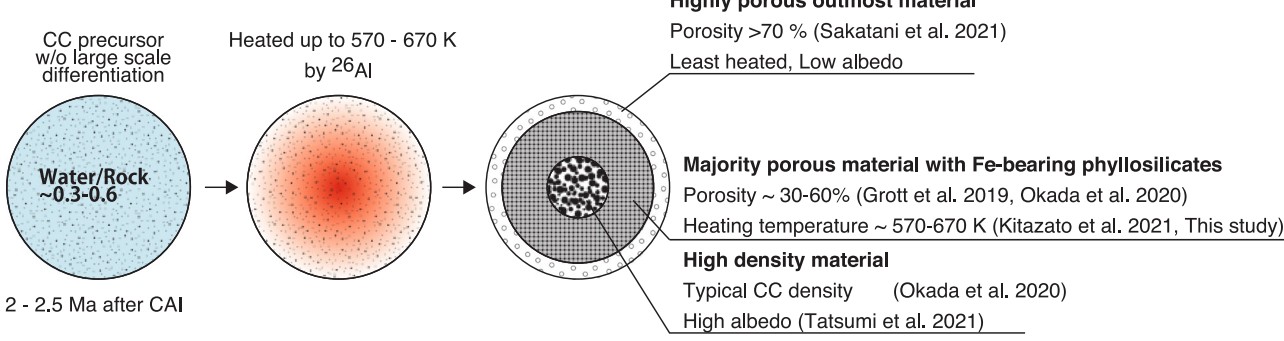

**Fig. 8 Hydrothermal history of Ryugu's parent body.** (Left) The parent body might consist of similar water/rock ratio to CM chondrites, but accreted with earlier timing, 2–2.5 Ma after CAI, to achieve the thermal metamorphism. (Middle) After pervasive aqueous alteration to form a CM-like precursor which included large amounts of Fe-bearing phyllosilicates, the parent body was dehydrated at 570–670 K (300–400 °C) due to heating by radioactive decay of short-lived radionuclides (e.g., $^{26}$Al). (Right) Ultimately, Ryugu's parent body consisted of several layers with different degrees of thermal metamorphism. Different evidence were discovered in the previous studies[5,15,41,56].

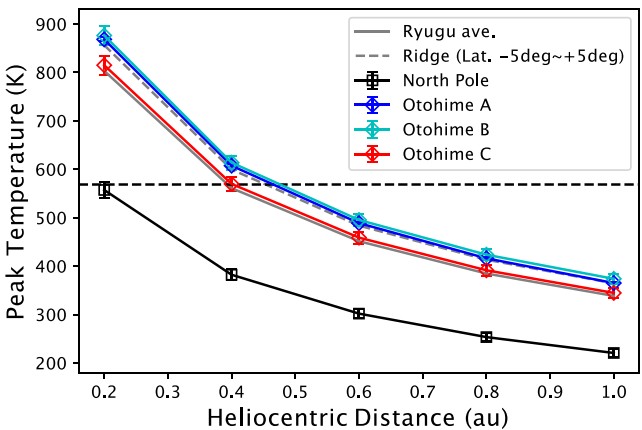

**Fig. 9 Diurnal peak temperature at equinox as a function of heliocentric distance.** North pole (black) and Otohime Facets (A: blue, B: cyan, C: red) are compared with the Ryugu average (gray solid line) and the equatorial ridge (gray dashed line). The equatorial ridge is defined as latitude from 5°S to 5°N. 570 K where phyllosilicates start to decompose is shown by black dashed horizontal line. North pole is kept lower than 570 K at 0.4 au, while Otohime Saxum is heated higher than 570 K.

0.48 μm, v: 0.55 μm, Na: 0.59 μm, w: 0.70 μm, x: 0.86 μm, p: 0.95 μm). Some images were contaminated by stray light from the radiator, which was strongly related to the attitude of the spacecraft with respect to the Solar radiation vector[12]. Because the stray light can be reproduced according to the spacecraft attitude, the stray light could be removed using the stray light template obtained by the same spacecraft attitude during cruising phase. After the stray light removal, other calibrations were conducted as same as the global dataset. Note that after the touchdowns (TDs) of Hayabusa2 on Ryugu, the optical efficiency of ONC-T was degraded by ~7% and 2–5% at v band, respectively[64]. Because the effect is not strongly dependent on the wavelength <1%, we could still evaluate the spatial distribution of spectral indices using normalized spectra obtained from the images after touchdowns[64]. There is a known issue that spectral sensitivity changes for the close encounter images, especially by scattered light inside of the hood of ONC-T. However, it is possible to evaluate spatial relative spectral variability. To obtain plausible spatial relative variations, we factored the images so that the median normalized I/F value in the images matched the I/F obtained by the global observation by ref.[3] and ref.[26]. After registration of images to the v-band image, the spectral indices, such as 0.7 μm-band absorption and b-to-x spectral slope, were calculated. The 0.7 μm-band absorption $d_w$ is calculated as

$$d_w = 1 - \frac{3.1 R_w}{1.6 R_v + 1.5 R_x} \quad (1)$$

where $R_n$ indicates the reflectance for the band filter and coefficients are wavelength weights. Reflectance values were weighted by the wavelength. When the value is positive, there is absorption at the w band. The ambiguity of absolute 0.7-μm absorption measurement from the sensitivity calibration of ONC-T is 1.6%[12].

This ambiguity is due to the spectral uncertainty of standard stars used for the radiometric calibration. However, spatial relative variation can be measured more accurately based on the onground and inflight signal-to-noise assessments. More specifically, the signal-to-noise ratio for one pixel is 200 for the reference temperature at −30 °C[12]. At the reference temperature, shot noise is the dominant factor and is random. Thus, we can reduce this random noise by 8 pixel by 8 pixel or 15 pixel by 15 pixel binnings to obtain better SNR, leading the detection capability of 0.7-μm to σ ~0.2% and σ ~0.1%, respectively, which is sufficient to evaluate the 0.3–0.7% spatial difference of 0.7-μm band absorption. To extract high accuracy spectra in Fig. 2, the average values were derived for the regions of interest (ROIs, see Supplementary Fig. 4), 15 pixels by 15 pixels square for each site. The error bars show the standard error inside of the ROIs.

**Near infrared spectroscopic analysis.** The near infrared spectrometer (NIRS3) also observed the south pole on 27 February 2019, 27 July 2019, and 26 October 2019, where all three facets of Otohime boulder were observed, and the north pole on 26 July 2019. For these observations, we extracted NIR spectra from both inside the region of interest (e.g., Otohime facets and the north polar region) and outside the region of interest as a standard for comparison. All four observations listed above have been acquired at different phases of the proximity phase (Supplementary Table 2). After 1st and 2nd TDs, the calibration standard, Radiometric Calibration Coefficient (RCC), had evolved, thus the standard reference spectrum had to be recalibrated for each observation. Also, for the observation of 27 February, the spacecraft trajectory was not completely reconstructed to provide robust observational information, such as incidence, and emission angles, and, as a result, no photometry correction was applied to this dataset. We show in Supplementary Fig. 5 that the photometry correction does not affect the band depth, as a result for relative comparison we applied non-photometric corrected data. The averaged reference spectra are calculated from a set of 100 spectra acquired on the same day, that fall outside of the region of interest and that present similar geometry of observations (similar incidence and emission angles).

**Temperature and photon dose calculation.** We calculated the average photon dose, as solar irradiation index, in the current orbit and pole orientation. We used the SPICE toolkit with SPICE kernels to calculate the values. The shape model constructed SHAPE_SFM_200k_v20200815 was used. The relative photon dose for each polygon at any specific time can be calculated as $\cos(i)/(D/1au)^2$, where $i$ is incidence angle with respect to the Sun and $D$ is distance between Ryugu and the Sun, (i.e., a polygon face perpendicular to the Sun at 1 au away has value 1). Depending on the season, solar photon dose distribution can vary due to the small obliquity of the rotational pole. Figure 4 shows the relative photon dose, i. e., solar irradiation, averaged in one asteroidal year.

To investigate the temperature that Ryugu's surface has been experiencing in its current orbit, we performed thermal simulation using the same global shape model. In this protocol, we solved a one-dimensional heat transfer equation for each facet on the shape model[65], with the following surface boundary condition,

$$I(1 - A) = -\Gamma \sqrt{\frac{\pi}{P}} \frac{\partial T}{\partial z}\bigg|_{z=0} + \varepsilon \sigma T_s^4 \quad (2)$$

where $T_s$ is surface temperature, $\Gamma$ is thermal inertia, $P$ is rotational period, $z$ is depth normalized by thermal skin depth, $A$ is albedo, and $I$ is input energy onto the surface that involves solar reflection and thermal radiation from the other facets (i.e., self-heating effect) as well as the direct solar irradiation. In our calculations, the thermophysical parameters of albedo, emissivity, and thermal inertia for each facet are set uniformly to 0.0146[66], 1.0, and 200 J m$^{-2}$ K$^{-1}$ s$^{-0.5}$ [35,42], respectively.

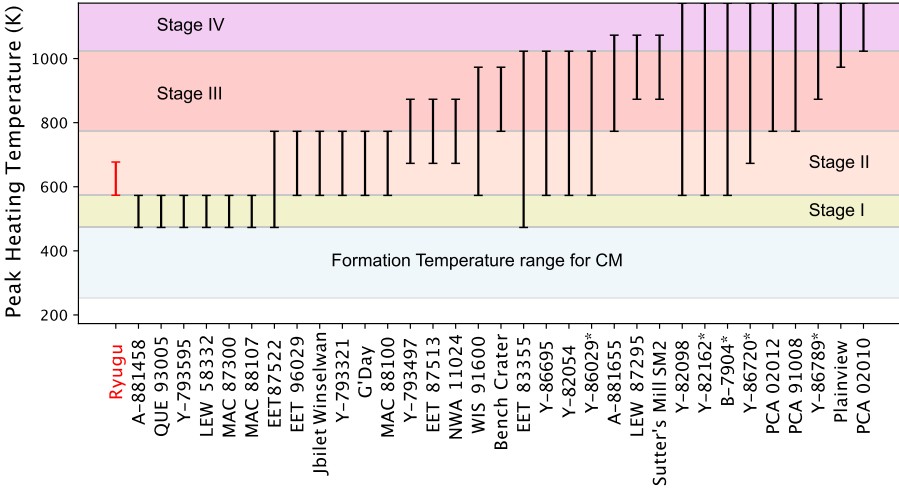

**Fig. 10 Estimated heating temperature for Ryugu and ATCCs.** Peak heating temperatures for meteorites were estimated by refs. [49, 50, 61]. The thermal metamorphism stages I–IV are defined by ref. [50]. Stage 1: low heating temperature of <570 K (<300 °C), Stage II: heating temperature of 570–770 K (300–500 °C), Stage III: heating temperature of 770–1020 K (500–750 °C), and Stage IV: high temperature of >1020 K (>750 °C). The meteorites with asterisks (*) are classified into CY, typically heated >770 K. Formation temperatures for CM chondrites were estimated by refs. [59–61].

The rotation period $P$ of Ryugu is set at 7.63 h. The ecliptic longitude and latitude of the rotational axis are given as 179.73 deg and −87.45 deg, respectively. The positions of Ryugu are set at the perihelion (ecliptic coordinate relative to the Sun of [−0.251, 0.928, −0.055] in astronomical unit) and the aphelion ([0.315, −1.379, 0.076]) in the current orbit of Ryugu. Supplementary Fig. 3 (a and b) shows the global distribution of the peak temperature at perihelion and aphelion. Supplementary Fig. 3c shows the diurnal peak temperature at the heliocentric distance of 0.2 au at equinox. Although the absolute value is different, the distribution of relatively colder regions is the same as the current orbit. In this case, most of the surface exceeds 570 K where the phyllosilicates start to be decomposed, while some regions around poles are kept under 570 K (Fig. 9). If Ryugu had experienced such close encounter to the Sun which was hypothesized in ref. [7], there could be large difference in 2.7-μm band absorption corresponding to phyllosilicate amount between polar regions and the typical Ryugu surface. Nevertheless, such a large difference was not observed (See Fig. 3).

We also reconstruct fields-of-view of ONC-T observations based on spacecraft attitudes and observation timings. We calculate temperature and photon dose for each pixel of each image (see also Supplementary Movies).

**Aqueously altered and thermally metamorphosed carbonaceous chondrites (ATCCs).** Carbonaceous chondrites that show evidence of aqueous alteration followed by thermal metamorphism were first termed as ATCCs by[43]. ATCCs including CYs (Yamato-type) experienced by a wide range of thermal metamorphism up to temperature of total dehydration and recrystallization at >1020 K. Several studies have estimated the heating temperature of ATCCs using X-ray diffraction and electron probe microanalysis[49,50,61], while Raman spectra of their insoluble organic matter has also been used to classify the degree of heating[63]. Figure 10 shows a summary of heating temperature of ATCCs from[49,50,61] compared with Ryugu's estimated heating temperature. Thermal metamorphism was classified into four stages by[50]. Stage I samples do not show heat-induced mineralogical changes in the X-ray diffraction patterns due to the low temperature <570 K (< 300 °C) they experienced. Stage II samples do not show either serpentine reflection nor secondary olivine and major phases include partially dehydrated amorphous phyllosilicates at the heating temperature of 570–770 K (300–500 °C). Stage III samples start to contain low-crystalline secondary olivine and no serpentine in the X-ray diffraction pattern at the heating temperature of 770–1020 K (500–750 °C). Fe-rich phyllosilicates are completely dehydrated at this stage. Stage IV samples are completely dehydrated and consist entirely of anhydrous minerals, such as recrystallized secondary olivine at the temperature >1020 K (>750 °C).

## Data availability
All Level 2d (I/F) images by ONC-T used in this study and derived data from images are available at https://data.darts.isas.jaxa.jp/pub/hayabusa2/paper/Tatsumi_2021/. The calibrated spectra by NIRS3 are available at Small Bodies Node of the NASA Planetary Data System (https://sbnarchive.psi.edu/pds4/hayabusa2/hyb2_nirs3/data_calibrated/proximity/). The input shape model, SPICE kernels and derived data supporting the findings of this study are available at https://data.darts.isas.jaxa.jp/pub/hayabusa2/paper/Tatsumi_2021/.

## Code availability
The code to calculate the asteroid's surface temperature is available from Naoya Sakatani (sakatani@rikkyo.ac.jp) on reasonable request. The code to calculate the photon dose on Ryugu is available at the JAXA DARTS (https://data.darts.isas.jaxa.jp/pub/hayabusa2/paper/Tatsumi_2021/).

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

## Acknowledgements

The Hayabusa2 spacecraft was developed and built under the leadership of JAXA, with contributions from DLR and CNES, and in collaboration with NASA, Nagoya Univ., Univ. of Tokyo, NAOJ, Univ. of Aizu, Kobe Univ. We also thank the engineers, Noriyasu Inaba of JAXA and Tetsuya Masuda, Seiji Yasuda, Kouta Matsushima, and Takeshi Ohshima of NEC Corp. for their work on the Hayabusa2 mission, Koshi Sato at NEC Corp. for ONC development, and Shingo Kashima at NAOJ for optical calculations. E.T. and S.S. were supported by KAKENHI from the JSPS Core-to-Core program "International Network of Planetary Sciences" and the Astrobiology Center Program of National Institutes of Natural Sciences (NINS) (Grant Number AB032004). T.H. acknowledges funding support from NASA's Emerging World/Planetary Data Archiving and Restoration. D.D. and F.V. acknowledge funding through the NASA's Hayabusa2 Participating Scientist Program (grant number NNX16AL34G), and NASA's Solar System Exploration Research Virtual Institute 2016 (SSERVI16) Cooperative Agreement (NNH16ZDA001N) for TREX (Toolbox for Research and Exploration). M.Z. was supported by NASA's Hayabusa2 Participating Scientist and Emerging Worlds Programs. E.P. received a support from the Italian Space Agency (ASI). M.P. acknowledges a grant of the Romanian National Authority for Scientific Research - UEFISCDI, project number PN-III-P1-1.1-TE-2019-1504.

## Author contributions

E.T. coordinated co-author contributions, led the ONC-T data analyses and interpretations. R.H., N.S., Y. Yokota, M. Yamada, S.S., T.M., S.K., H. Sawada., M.M., T.K., E.T., C.H., K.O., H.Suzuki., M.I., K.Y., M.H., Y.C., C.S., and M.S. performed the ONC-T data acquisitions and reductions. L.R., C.P., and K.K. performed the NIRS3 data acquisitions and reductions. N.S., S.T., Y.S., and J.T. carried out the thermal modelling. Nao. Hirata, Nar. Hirata, and Y. Yamamoto. contributed for the shape modelling and the estimation of spacecraft trajectory. S.T., M.H., Y. Tsuda, M. Yoshikawa, T.S., F.T., S.N., Y. Takei, H.T., Y. Yamamoto, H. Sawada., and T.O. carried out the science operations of spacecraft. Y. Tsuda and S.N. carried out the project general management. S.W. and M. Yoshikawa carried out the project science management. S.T. developed and operated TIR. N.N. developed and operated LIDAR. E.T., S.S., N.S., L.R., M.M., R.H., T.M., T.N., M.Z., R.B., T.H., S.S., S.W., J.d.L., M.P., J.L.R.G., J.L., E.P., D.D., F.V., and H.C. contributed for interpretation and writing the manuscript.

## Competing interests

The authors declare no competing interests.
