## [Peer Review File · Nature Communications]

REVIEWER COMMENTS

Reviewer #1 (Remarks to the Author):

This is my second review of this manuscript where they reports visible and near-infrared spectra of the polar regions of the asteroid Ryugu. The measurements were performed by the Hayabusa2 space mission during close flybys. The authors use the data to discuss the aqueous and thermal history of Ryugu. I think the authors did a good job for addressing my previous comments. I still have few comments and questions that should be addressed before the paper can be accepted.

1- You should not capitalize solar system according to Nasa (Capitalize the names of planets (e.g., "Earth," "Mars," "Jupiter"). Capitalize "Moon" when referring to Earth's Moon; otherwise, lowercase "moon" (e.g., "The Moon orbits Earth," "Jupiter's moons"). Capitalize "Sun" when referring to our Sun but not to other suns. Do not capitalize "solar system" and "universe.")

2- The sentence lines 285 to 289 is extremely confusing and should be rephrased.

3- Lines 308-312. Does the 0.7- μm band would be observed by the spacecraft at a very low level of < 2%? If not, it can put constraint on the maximum temperature experienced by Ryugu, which should be ≤ 700 K. The detection limit of the 0.7- μm and the implication on the maximum temperature should be discussed.

4- Lines 320-322. Could D-type asteroids represent the potential unsampled hydrated carbonaceous chondrites? Band adsorption might be available for Tagish Lake. In addition, recent paper suggest that Tarda could represent a D-type-derived sample (Marrocchi + 2021, ApJ Letters). Should be discussed.

5- What about CY chondrites? They experiences hydration and dehydration and could represent a good analogue (see the work by King + 2019 GCA).

5- I think the authors should discussed the temperature experienced by regular CM chondrites. For instance, temperatures up to 550 K have been proposed for CMs (Verdier-Paoletti+ 2017 EPSL) and are this closed to the 700 K proposed for Ryugu. In a general manner, the CMs likely record higher temperatures than the authors claimed (< 423 K, line 346). It should be discussed in the light of

recent papers that estimated the CM temperatures (Verdier-Paoletti + 2017, EPSL; Vacher+ 2019a ApJ Letters, 2019b, MAPS; King+ 2021 GCA; Velbel & Zolensky 2021 MAPS).

King, A.J., Bates, H.C., Krietsch, D., Busemann, H., Clay, P.L., Schofield, P.F., Russell, S.S., 2019. The Yamato-type (CY) carbonaceous chondrite group: Analogues for the surface of asteroid Ryugu? *Geochemistry* 79, 125531. <https://doi.org/10.1016/j.chemer.2019.08.003>

King, A.J., Schofield, P.F., Russell, S.S., 2021. Thermal alteration of CM carbonaceous chondrites: Mineralogical changes and metamorphic temperatures. *Geochimica et Cosmochimica Acta* 298, 167–190. <https://doi.org/10.1016/j.gca.2021.02.011>

Marrocchi, Y., Avice, G., Barrat, J.-A., 2021. The Tarda Meteorite: A Window into the Formation of D-type Asteroids. *The Astrophysical Journal Letters* L9.

Vacher, L.G., Piralla, M., Gounelle, M., Bizzarro, M., Marrocchi, Y., 2019a. Thermal Evolution of Hydrated Asteroids Inferred from Oxygen Isotopes. *The Astrophysical Journal Letters* 882, 0–0. <https://doi.org/10.3847/2041-8213/ab3bd0>

Vacher, L.G., Truche, L., Faure, F., Tissandier, L., Mosser-Ruck, R., Marrocchi, Y., 2019b. Deciphering the conditions of tochilinite and cronstedtite formation in CM chondrites from low temperature hydrothermal experiments. *Meteorit Planet. Sci.* 54, 1870–1889. <https://doi.org/10.1111/maps.13317>

Velbel, M.A., Zolensky, M.E., 2021. Thermal metamorphism of CM chondrites: A dehydroxylation-based peak-temperature thermometer and implications for sample return from asteroids Ryugu and Bennu. *Meteorit Planet Sci* 56, 546–585. <https://doi.org/10.1111/maps.13636>

Verdier-Paoletti, M.J., Marrocchi, Y., Avice, G., Roskosz, M., Gurenko, A., Gounelle, M., 2017. Oxygen isotope constraints on the alteration temperatures of CM chondrites. *Earth and Planetary Science Letters* 458, 273–281. <https://doi.org/10.1016/j.epsl.2016.10.055>

Reviewer #2 (Remarks to the Author):

I thank the authors for their detailed responses. It is clear that the authors made a good-faith effort to address the comments and concerns of both reviewers. I am satisfied with the responses and revisions and have only one additional comment that I'd like to see addressed before publication:

Lines 211-216. The text in this section contains several unsupported assumptions. The first is that the optical effects of space weathering due to solar wind irradiation occur along a linear continuum such that a region that experiences 20% of the solar wind irradiation of another region will take 5 times longer to register the same optical effects. The second relates to the rate of space weathering and that Facet A is young enough and has undergone so little space weathering that it represents the original B-type color of Ryugu. Supporting evidence and references for these assumptions are necessary to include here.

Review by Tim Glotch, Stony Brook University

We thank both reviewers for a close look at our paper and insightful suggestions. We agree with the comments and update our manuscript. The updated parts are shown using the Microsoft Word review function.

REVIEWER COMMENTS

Reviewer #1 (Remarks to the Author):

This is my second review of this manuscript where they reports visible and near-infrared spectra of the polar regions of the asteroid Ryugu. The measurements were performed by the Hayabusa2 space mission during close flybys. The authors use the data to discuss the aqueous and thermal history of Ryugu. I think the authors did a good job for addressing my previous comments. I still have few comments and questions that should be addressed before the paper can be accepted.

1- You should not capitalize solar system according to Nasa (Capitalize the names of planets (e.g., "Earth," "Mars," "Jupiter"). Capitalize "Moon" when referring to Earth's Moon; otherwise, lowercase "moon" (e.g., "The Moon orbits Earth," "Jupiter's moons"). Capitalize "Sun" when referring to our Sun but not to other suns. Do not capitalize "solar system" and "universe.")

Thank you very much. I did not know this. I corrected it.

2- The sentence lines 285 to 289 is extremely confusing and should be rephrased.

Thank you for suggestion. We re-phrased those sentences to be clear.

3- Lines 308-312. Does the 0.7- μm band would be observed by the spacecraft at a very low level of < 2%? If not, it can put constraint on the maximum temperature experienced by Ryugu, which should be ≤ 700 K. The detection limit of the 0.7- μm and the implication on the maximum temperature should be discussed.

This is very important point. As we described in the manuscript, the absolute measurement of 0.7- μm depth has a large error 1.6% compared with its shallow absorption. This is mainly because of the radiometric calibration accuracy. the ambiguity in the sensitivities for the v-, w-, and x-bands are 0.85%, 1.3%, and 1.6%, respectively. Thus, our estimation of absolute depth of 0.7- μm band depth is 0 to 3%. We agree that 0.7- μm band can be lost by decomposing of Fe-bearing phyllosilicate at 570-670 K (Hiroi et al. 1996 MAPS, Cloutis et al. 2012 Icarus). This can be stronger constrain than elimination of 2.7- μm at 770 – 970 K. We changed the upper limit of heating temperature to 670 K according to this.

4- Lines 320-322. Could D-type asteroids represent the potential unsampled hydrated carbonaceous chondrites? Band adsorption might be available for Tagish Lake. In addition, recent paper suggest that Tarda could represent a D-type-derived sample (Marrocchi + 2021, ApJ Letters). Should be discussed.

Thank you for the comment. Actually we also had considered about Tagish Lake as a possible similar material to Ryugu. 0.7- μm and 2.7- μm band absorption of Tagish Lake was measured in Hiroi et al. (2001). However, we don't think Tagish Lake represents Ryugu well because of following reasons. It shows sharp 2.7- μm band but no 0.7- μm band. Moreover, the visible spectra of Tagish Lake is particularly red, which is inconsistent with Ryugu's flat spectrum. Another point is that Tagish Lake may become bluer by space weathering (Trang et al. 2021), which is also inconsistent with what we found on Ryugu. Tagish Lake could represent D-type asteroids which are very red but not B or C

types. Marrocchi et al. (2021) shows the compositional similarity between Tagish Lake and Tarda, but there are no spectra available for comparing with our remote-sensing results so far. Thus, it is difficult to say if Tarda is similar to Ryugu or not. But definitely returned samples can tell us the similarity with those meteorites.

5- What about CY chondrites? They experience hydration and dehydration and could represent a good analogue (see the work by King + 2019 GCA).

Yes, CYs may be compositionally similar to Ryugu and we don't rule out CYs as Ryugu analogs. Actually ATCCs include CY chondrites, but the heating temperature is relatively high, >770 K, so that the Fe-bearing phyllosilicates are dehydrated. Moreover, we have not found exact match with the CY spectra. Thus, we don't think CYs are not the best candidate, but there could be a relation. We pointed out the difference in hydration between Ryugu and CYs in the text.

5- I think the authors should discuss the temperature experienced by regular CM chondrites. For instance, temperatures up to 550 K have been proposed for CMs (Verdier-Paoletti+ 2017 EPSL) and are thus close to the 700 K proposed for Ryugu. In a general manner, the CMs likely record higher temperatures than the authors claimed (< 423 K, line 346). It should be discussed in the light of recent papers that estimated the CM temperatures (Verdier-Paoletti + 2017, EPSL; Vacher+ 2019a ApJ Letters, 2019b, MAPS; King+ 2021 GCA; Velbel & Zolensky 2021 MAPS).

Thank you very much for another insightful suggestion. Actually I learned a lot from those recent publications. We added discussions on formation temperature and heating temperature of ATCCs. As you suggested, the range gets very close between CM formation temperature and heating temperature of ATCCs. We found this is very important and implemented in the discussion and methods.

Reviewer #2 (Remarks to the Author):

I thank the authors for their detailed responses. It is clear that the authors made a good-faith effort to address the comments and concerns of both reviewers. I am satisfied with the responses and revisions and have only one additional comment that I'd like to see addressed before publication:

Lines 211-216. The text in this section contains several unsupported assumptions. The first is that the optical effects of space weathering due to solar wind irradiation occur along a linear continuum such that a region that experiences 20% of the solar wind irradiation of another region will take 5 times longer to register the same optical effects. The second relates to the rate of space weathering and that Facet A is young enough and has undergone so little space weathering that it represents the original B-type color of Ryugu. Supporting evidence and references for these assumptions are necessary to include here.

Review by Tim Glotch, Stony Brook University

Thank you for pointing out this a very important point.

As you may know, the optical effects of space weathering are not linear, especially well understood with ordinary chondrites (e.g., Brunetto and Strazzulla, 2005) and some more recent results for carbonaceous chondrites (Lantz et al. 2017). More specifically, the slope variations we saw on both OCs and CCs with respect to accumulated ion fluence (proportional to the exposure time in space) tend to be exponential with asymptotic values (e.g., $\text{slope} = a + b \cdot \exp(\text{Fluence} \cdot c)$), same for albedo variations. It means the optical effects of space weathering slows down with aging. Actually, we can see such non-linear trend in our data too in Figure 5. The dose is proportional to the exposure time, we can convert photon dose to relative exposure time of the surface. We added secondary x axis in Fig. 5 (a) and (b). We also added discussion in the text.

REVIEWERS' COMMENTS

Reviewer #1 (Remarks to the Author):

The authors did a good job and they responded specifically to my last comments. I am happy to accept the article.

Best regards

Reviewer #2 (Remarks to the Author):

The authors have adequately addressed all remaining questions. I recommend publication of the manuscript in its current form.